

# Mangrove afforestation as an ecological control of invasive *Spartina alterniflora* affects rhizosphere soil physicochemical properties and bacterial community in a subtropical tidal estuarine wetland

Jinwang Wang[1], Xi Lin[2], Xia An[3], Shuangshuang Liu[1], Xin Wei[1], Tianpei Zhou[4], Qianchen Li[5], Qiuxia Chen[1] and Xing Liu[1]

[1] Wenzhou Key Laboratory of Resource Plant Innovation and Utilization, Zhejiang Institute of Subtropical Crops, Zhejiang Academy of Agricultural Sciences, Wenzhou, China
[2] Wenzhou Institute of Eco-Environmental Sciences, Wenzhou, China
[3] Zhejiang Xiaoshan Institute of Cotton & Bast Fiber Crops, Zhejiang Institute of Landscape Plants and Flowers, Zhejiang Academy of Agricultural Sciences, Hangzhou, China
[4] Yueqing Bureau of Natural Resources and Planning, Wenzhou, China
[5] College of Life and Environmental Sciences, Wenzhou University, Wenzhou, China

Corresponding authors
Qiuxia Chen, yzscqx@163.com
Xing Liu, liuxingzy1989@163.com

## ABSTRACT

**Background:** The planting of mangroves is extensively used to control the invasive plant *Spartina alterniflora* in coastal wetlands. Different plant species release diverse sets of small organic compounds that affect rhizosphere conditions and support high levels of microbial activity. The root-associated microbial community is crucial for plant health and soil nutrient cycling, and for maintaining the stability of the wetland ecosystem.

**Methods:** High-throughput sequencing was used to assess the structure and function of the soil bacterial communities in mudflat soil and in the rhizosphere soils of *S. alterniflora*, mangroves, and native plants in the Oujiang estuarine wetland, China. A distance-based redundancy analysis (based on Bray–Curtis metrics) was used to identify key soil factors driving bacterial community structure.

**Results:** *S. alterniflora* invasion and subsequent mangrove afforestation led to the formation of distinct bacterial communities. The main soil factors driving the structure of bacterial communities were electrical conductivity (EC), available potassium (AK), available phosphorus (AP), and organic matter (OM). *S. alterniflora* obviously increased EC, OM, available nitrogen (AN), and $NO_3^-$-N contents, and consequently attracted copiotrophic Bacteroidates to conduct invasion in the coastal areas. Mangroves, especially *Kandelia obovata*, were suitable pioneer species for restoration and recruited beneficial Desulfobacterota and Bacilli to the rhizosphere. These conditions ultimately increased the contents of AP, available sulfur (AS), and AN in soil. The native plant species *Carex scabrifolia* and *Suaeda glauca* affected coastal saline soil primarily by decreasing the EC, rather than by increasing nutrient contents. The predicted functions of bacterial communities in rhizosphere soils were related to active catabolism, whereas those of the bacterial community in mudflat soil were related to synthesis and resistance to environmental factors.

**Conclusions:** Ecological restoration using *K. obovata* has effectively improved a degraded coastal wetland mainly through increasing phosphorus availability and promoting the succession of the microbial community.

## INTRODUCTION

Coastal wetlands are among the most valuable ecosystems on Earth and have important ecological and commercial functions, such as supporting biodiversity, providing food, and remediating pollutants (*Liu & Ma, 2024*). *Spartina alterniflora*, the most successful invasive species worldwide, has rapidly spread in Chinese coastal wetlands since 1979, and now covers a total area of ~67,500 ha (*Gu et al., 2021*). Previous studies have reported that *S. alterniflora* invasion could severely decrease the biodiversity by excluding native plant species, and consequently altered processes and functions of coastal wetland ecosystems (*Lin et al., 2022*; *Yang et al., 2020*; *Zhang et al., 2010*; *Zheng et al., 2023*). Therefore, there is an urgent need to control invasive *S. alterniflora* using ecological measures, such as plant transplantation and seeding, to increase the resilience of biodiversity and repair ecosystem services (*Liu & Ma, 2024*). As the only woody halophytes that grow in tidal zones along tropical and subtropical coastlines, mangroves provide immense ecological services, such as biodiversity conservation, coastal protection, and nutrient retention (*Friess et al., 2019*). Thus, planting mangroves is considered an effective and appropriate method of ecological restoration for coastal wetlands (*Zhao et al., 2016*). Mangroves are a particularly vulnerable group of temperature-sensitive plants, with a narrow geographical distribution ranging from Sanya (18°12′N) to Fuding (27°20′N) in China (*Liu et al., 2022a*). In response to global warming, the distribution of mangroves may be pushed to their latitudinal limits. Consequently, the government of Zhejiang Province (27°02′–31°11′N) has initiated a series of mangrove afforestation projects. In particular, mono-specific stands of *Kandelia obovata* covering large areas (~500 ha) have been established.

Soil property and plant diversity and productivity co-regulate the variation and succession of soil microbial communities (*Wu et al., 2018*). At the same time, microorganisms constitute an interactive network between plants and soil (*Nadarajah & Abdul Rahman, 2021*). Even minor changes in the microbial community can significantly affect plant growth, soil nutrient cycling, and ecosystem multifunctionality. Several recent studies have focused on the characteristics of the microbial community and its roles in regulating biogeochemical processes in mangrove ecosystems invaded by *S. alterniflora* (*Gao et al., 2019*; *Lin et al., 2022*; *Liu et al., 2017*; *Zheng et al., 2017*). *Liu et al. (2017)* found that the alpha diversity of soil bacteria markedly increased in natural mangrove stands after *S. alterniflora* invasion, while *Gao et al. (2019)* detected the opposite trend. Compared with *Phragmites australis* or *K. obovata*, invasive *S. alterniflora* more strongly affected bacterial community assembly processes (*Lin et al., 2022*). *S. alterniflora* was found to

affect soil nitrogen (N) and sulfate (S) cycles by regulating denitrification and S reduction processes (*Gao et al., 2019*; *Zheng et al., 2017*). Notably, microbial communities are regulated by habitat-specific biotic and abiotic factors. To date, little is known about the changes in microbial ecology (*e.g.*, bacterial community structure and its potential functions) and soil nutrient dynamics in response to *S. alterniflora* invasion or mangrove afforestation at higher latitudes (*e.g.*, Zhejiang Province) beyond the northernmost limit of the natural distribution of mangroves.

The rhizosphere harbors high microbial diversity and activity because of the presence of root exudates and a reliable oxygen supply. This high level of diversity positively contributes to soil nutrient cycling, plant growth, and ecosystem productivity (*Pii et al., 2015*). In addition, *Lu et al. (2022)* detected a closer relationship among bacterial species in mangrove rhizosphere soil than in adjacent mudflat soil. Bacteria, the most abundant and diverse group of organisms on earth, play vital roles in regulating nearly all biogeochemical processes (*e.g.*, N fixation, organic matter (OM) degradation, phosphate (P) solubilization, and S reduction). Therefore, the bacterial community is of great importance for maintaining the health and stability of mangrove ecosystems (*Lin et al., 2019*; *Lu et al., 2022*; *Thatoi et al., 2013*). To the best of our knowledge, no previous studies have explored the potential opportunities and/or risks to rhizosphere soil bacterial communities when mangrove afforestation is used as a strategy to control invasive *S. alterniflora* in subtropical tidal estuarine wetlands >27°20′N. Moreover, because mangroves and *S. alterniflora* are introduced species, it is important to determine which plant has greater effects on local bacterial communities, and to identify the key environmental factors driving variations in bacterial community structure. To fill this knowledge gap, we investigated the characteristics of the bacterial communities, and their succession mechanisms, in rhizosphere soils of three types of vegetation: native wetland plants (*Suaeda glauca* and *Carex scabrifolia*), the mangroves (*K. obovata* and *Sonneratia apetala*), and the invasive plant *S. alterniflora*, in a subtropical estuarine wetland, using adjacent mudflat soil as the control. Our main objectives were as follows: (1) to determine bacterial community structure and function, as well as soil physicochemical properties, in the rhizosphere of native plants, and to explore how these bacterial communities respond to *S. alterniflora* invasion and mangrove afforestation; and (2) to explore key soil factors driving bacterial community succession in a coastal wetland. The results of this study provide new insights into the effects of *S. alterniflora* invasion and mangrove afforestation on coastal ecosystems, which will be useful for designing and implementing afforestation projects for coastal restoration.

# MATERIALS AND METHODS

## Study area

This field experiment was carried out in a coastal wetland (120°51′–120°52′E, 27°56′–27°57′N) on Shupaisha Island, in a branch of the Oujiang estuary, Wenzhou City, Zhejiang Province, China. This area is under a typical subtropical monsoon climate, and the mean annual air temperature is 18.04 °C. The tidal regime is characterized by irregular

semidiurnal tides with a mean tidal height range of 4.5 m. The soil originates from modern marine and fluvial deposits with a high percentage of exchangeable $Na^+$ and low N and P contents. According to the USDA classification, the soil is classified as an Inceptisol (*Sahoo et al., 2023*), with a silty clay loam texture (74.6 ± 7.8% silt, 31.4 ± 3.1% clay). The main native plants in this wetland were *S. glauca*, *C. scabrifolia*, *P. australis*, and *Scirpus × mariqueter*. Cordgrass (*S. alterniflora*) was introduced in 1989, and has since spread aggressively and occupied the environment by excluding native vegetation, ultimately becoming the dominant plant in the wetland. Since 2014, seedlings of various mangrove species, including *K. obovata*, *S. apetala*, *Aegiceras corniculatum* and *Excoecaria agallocha*, have been successively transplanted for ecological restoration of this damaged coastal wetland. *K. obovata*, the most cold-resistant mangrove species, is the only tree species recommended for afforestation in Zhejiang Province, and it now covers more than 90% of Shupaisha Island. The coexistence of mangroves and *S. alterniflora* on Shupaisha Island meant that the site was suitable for research on the effects of *S. alterniflora* invasion and mangrove introduction on a subtropical tidal estuarine wetland.

## Experimental sampling and processing of samples

Six sampling sites were selected, and included three vegetation types: (i) native wetland plants (*S. glauca* and *C. scabrifolia*); (ii) mangroves (*K. obovata* and *S. apetala*); and (iii) the invasive plant *S. alterniflora*. Mudflat soil was collected as the control. *S. glauca* is an annual $C_3$ grass with a single primary root and undeveloped fibrous roots. It often grows in saline soil environments such as seashores and wastelands, as well as in the high-tide zone or supratidal zone. *C. scabrifolia*, a perennial $C_3$ grass with fine fibrous roots and underground stolons, is usually found in coastal mudflats in the mid-high-tide zone or supratidal zone and constitutes a distinctive grass felt layer. *K. obovata*, a $C_3$ woody plant with a taproot system and a well-developed root system, is usually found on muddy high-tidal mudflats with deep silt alluvium in bays. *S. apetala*, a $C_3$ woody plant with a taproot system and a respiratory root, usually grows on muddy high-tidal mudflats with deep silt alluvium in bays. *S. alterniflora*, an exotic $C_4$ perennial grass with short, thin fibrous roots and rhizomes, usually grows in the intertidal zone of estuaries, bays, and other coastal mudflats, and often forms dense single-species communities with a distinctive grass felt layer. The mudflat was selected as the background ecosystem prior to the invasion of *S. alterniflora* or the introduction of mangroves. In August 2020, three 20 m × 20 m plots were randomly established to sample the five plant species and the bare tidal mudflat on Shupaisha Island. In each plot, the rhizosphere soil was brushed off from fibrous roots from three to five plants or bushes and then mixed to form one representative sample. Five soil cores were randomly collected from 0–20 cm depths using a stainless-steel auger (diameter, 5 cm) in each mudflat plot. All composite soil samples were divided into two portions. One portion was freeze-dried and then stored at −80 °C until DNA extraction, and the other portion was air-dried and sieved through a 2-mm mesh for soil physicochemical analyses.

## Soil property measurements

Soil pH and electrical conductivity (EC) were measured in a soil:deionized water mixture (1:5, w/w) using a pH meter (Metler-Toledo, Greisensee, Switzerland) and a conductivity meter (Leici, Shanghai, China), respectively. The soil organic matter (OM) content was determined by potassium dichromate oxidation with external heating. Soil available nitrogen (AN), $NH_4^+$-N, and $NO_3^-$-N contents were determined by alkaline hydrolysis diffusion, indophenol blue colorimetry, and phenol disulfonic acid methods, respectively. Soil available phosphorus (AP) was extracted using 0.5 mol $L^{-1}$ $NaHCO_3$ and measured using the Mo-Sb anti-spectrophotometric method. The soil available potassium (AK) content was determined by flame photometry. Available sulfur (AS) was extracted from soil using $CaCl_2$ and then quantified using the barium sulfate turbidimetric method.

## DNA extraction, PCR amplification, high-throughput sequencing, and data processing

Total genomic DNA was extracted from soil samples using a PowerSoil DNA Isolation Kit (Mo Bio Laboratories, Carlsbad, CA, USA) following the manufacturer's protocol. The quantity and quality of extracted DNA was evaluated using a NanoDrop® ND-2000 spectrophotometer (Thermo Scientific Inc., Waltham, MA, USA) and by gel electrophoresis (2.0% w/v agarose). According to a previous study (*Song et al., 2022*), bacterial 16S RNA genes were amplified by polymerase chain reaction (95 °C for 3 min followed by 27 cycles of denaturation at 95 °C for 30 s, annealing at 55 °C for 30 s, extension at 72 °C for 45 s, with final extension at 72 °C for 10 min) using the primers 338F (5′-ACTCCTACGGGAGGCAGCAG-3′) and 806R (5′-GGACTACHVGGGTWTCTAAT-3′) (ABI GeneAmp 9700, Life Technologies, Carlsbad, CA, USA). Each 20-μL PCR mixture consisted of 4 μL 5 × Fast Pfu buffer, 2 μL 2.5 mM dNTPs, 0.8 μL each primer (5 μM), 0.4 μL Fast Pfu polymerase, 10 ng template DNA, and ddH$_2$O. All samples were amplified in triplicate. The PCR products were detected using 2.0% w/v agarose gel electrophoresis and then purified using an AxyPrep DNA Gel Extraction Kit (Axygen Biosciences, Union City, CA, USA). Finally, the purified amplicons were pooled in equimolar amounts, and then paired-end sequencing (2 × 300) was carried out on the MiSeq PE300 platform (Illumina, San Diego, CA, USA) at the Centre for Genomic Research, Shanghai Majorbio Biotechnology Co. Ltd. (Shanghai, China).

Raw sequencing reads were de-multiplexed using USEARCH 11 (https://drive5.com/usearch/) and filtered using fastp v0.19.6. The reads with average quality score <20 or containing consecutive "N" bases were discarded. Using FLASH v1.2.7, clean reads longer than 50 bp were then merged (maximum mismatch rate of the overlap region, 0.1), and only sequences longer than 200 bp were retained as previously described (*Liu et al., 2022b*). The resulting sequences were clustered into operational taxonomic units (OTUs) at 97% similarity using the UPARSE algorithm, and chloroplast sequences were removed. The number of 16S rRNA gene sequences from each sample was rarefied to 41,216, which yielded an average Good's coverage of ~97%. Taxonomic labels were assigned to OTUs

using the Ribosomal Database Project (RDP, https://sourceforge.net/projects/rdp-classifier/) Bayesian classifier, with a confidence threshold of 70%, at the SILVA reference database (ver. 138). Based on OTU representative sequences, metagenomic functions were predicted using PICRUSt2 (Phylogenetic Investigation of Communities by Reconstruction of Unobserved States), as described in detail elsewhere (*Douglas et al., 2020*)

## Data analysis

Data are expressed as average ± standard error of three biological replicates. The data were subjected to homogeneity of variance and normality tests to determine their suitability for ANOVA. The statistical analyses were conducted using SPSS 21.0 and R-3.3.1 software. Duncan's t-test was used to detect differences in soil physicochemical properties among samples. Bacterial community composition and 16S predicted functional profiles were analyzed using the free online platform I-Sanger (https://www.majorbio.com/). The community richness indexes (Sobs and Ace) and community diversity indexes (Shannon's and Simpson's) were calculated with Mothur v1.30.1 and analyzed by one-way ANOVA, followed by Tukey's test for multiple comparisons. Based on Bray–Curtis dissimilarity, the similarity among bacterial communities was analyzed by principal coordinate analysis (PCA). A PERMANOVA test was used to determine the percentage of variation explained by the soil type, and its statistical significance, using the Vegan v2.5-3 package. Linear discriminant analysis (LDA) effect size (LEfSe) (http://huttenhower.sph.harvard.edu/LEfSe) was used to identify the significant differentially abundant taxa (at phylum to genus levels) among different soil samples (LDA score > 4, $P < 0.05$). Venn diagrams were used to illustrate the shared and unique taxa in different soil samples. To account for multicollinearity among the nine soil properties, the variance inflation factor (VIF) for each variable was estimated as previously described (*Lin et al., 2022*). The relationships between soil physicochemical properties and bacterial community structure were determined by redundancy analysis (RDA) using the Vegan v2.5-3 package. A Procrustes analysis was conducted to compare the degree of fit between the spatial ranking of soil characteristics and bacterial community structure.

## RESULTS

### Soil physicochemical properties

As shown in Table 1, *S. alterniflora* invasion increased the EC of rhizosphere soil. The EC value of *S. alterniflora* rhizosphere soil was 7.25 ± 0.15 dS m$^{-1}$, whereas that of mudflat soil was 6.55 ± 0.16 dS m$^{-1}$ ($P < 0.05$). In contrast, the EC value was 5.8–44.7% lower in other rhizosphere soil samples than in mudflat soil, with values ranging from 3.74 ± 0.23 to 6.16 ± 0.16 dS m$^{-1}$ ($P < 0.05$). The soil pH was highest in the rhizosphere soil of *S. glauca* (8.39 ± 0.09) and lowest in the rhizosphere soil of *S. alterniflora* (7.98 ± 0.09), but not significantly different among the other soil samples (range, 8.24 ± 0.05 to 8.28 ± 0.06; $P > 0.05$). Except for *S. apetala* and *C. scabrifolia*, the presence of vegetation resulted in substantially higher OM contents in the rhizosphere soil than in mudflat soil ($P < 0.05$). Specifically, the soil OM content was highest in the rhizosphere soil of *S. alterniflora* (18.07 ± 0.71 g kg$^{-1}$), followed by the rhizosphere soils of *K. obovata* (16.87 ± 0.45 g kg$^{-1}$) and *S. glauca* (16.27 ±

**Table 1 Soil physiochemical properties.**

| Site | pH | EC (ds m$^{-1}$) | OM (g kg$^{-1}$) | AN (mg kg$^{-1}$) | NH$_4^+$-N (mg kg$^{-1}$) | NO$_3^-$-N (mg kg$^{-1}$) | AP (mg kg$^{-1}$) | AK (mg kg$^{-1}$) | AS (mg kg$^{-1}$) |
|------|-----|------|------|------|------|------|------|------|------|
| Mud | 8.27 ± 0.08b | 6.55 ± 0.16b | 14.73 ± 0.40c | 48.74 ± 1.61cd | 36.27 ± 1.15b | 6.98 ± 0.13d | 5.74 ± 0. 30cd | 377.7 ± 15.4a | 184.3 ± 4.0c |
| KO | 8.12 ± 0.08cd | 6.16 ± 0.16c | 16.87 ± 0.45b | 52.99 ± 1.28b | 37.80 ± 1.42b | 7.47 ± 0.14c | 11.77 ± 1.48a | 359.7 ± 21.2a | 217.7 ± 5.5a |
| SO | 8.24 ± 0.05bc | 4.08 ± 0.16d | 14.83 ± 0.57c | 49.76 ± 1.13c | 35.17 ± 1.79bc | 6.79 ± 0.23d | 7.16 ± 0.25b | 324.7 ± 21.2b | 156.0 ± 9.2e |
| SG | 8.39 ± 0.09a | 4.31 ± 0.20d | 16.27 ± 0.50b | 51.01 ± 1.96bc | 36.97 ± 1.51b | 8.89 ± 0.31b | 5.14 ± 0.41d | 320.0 ± 20.9b | 166.7 ± 3.0d |
| CT | 8.28 ± 0.06ab | 3.74 ± 0.23e | 14.64 ± 0.38c | 46.43 ± 0.94d | 42.90 ± 1.61a | 2.80 ± 0.30e | 6.41 ± 0.32bc | 363.7 ± 19.4a | 133.7 ± 6.1f |
| SA | 7.98 ± 0.09d | 7.25 ± 0.15a | 18.07 ± 0.71a | 58.11 ± 1.64a | 33.23 ± 1.80c | 14.99 ± 0.26a | 4.73 ± 0.11d | 307.0 ± 21.1b | 204.3 ± 5.1b |

**Note:**
Mud, bare mudflat; KO, *Kandelia obovata*; SO, *Sonneratia apetala*; SG, *Suaeda glauca*; CT, *Carex scabrifolia*; SA, *Spartina alterniflora*; EC, electrical conductivity; OM, organic matter; AN, available nitrogen; NH$_4^+$-N, ammonium nitrogen; NO$_3^-$-N, nitrate nitrogen; AP, available phosphate; AK, available potassium; AS, available sulfur. Different lowercase letters indicated the significant differences among different vegetation types at $P < 0.05$ according to the LSD test.

0.50 g kg$^{-1}$). Similarly, the AN and NO$_3^-$-N contents were lower in mudflat soil (48.74 ± 1.61 mg kg$^{-1}$ and 6.98 ± 0.13 mg kg$^{-1}$, respectively) than in *S. alterniflora* rhizosphere soil (58.11 ± 1.64 and 14.99 ± 0.26 mg kg$^{-1}$, respectively) ($P < 0.05$). The NH$_4^+$-N content was highest in *C. scabrifolia* rhizosphere soil and lowest in *S. alterniflora* rhizosphere soil, but not significantly different among the other rhizosphere soil samples and mudflat soil ($P > 0.05$). Notably, the AP content was markedly higher in the rhizosphere soils of the *K. obovata* (11.77 ± 1.48 mg kg$^{-1}$) and *S. apetala* (7.16 ± 0.25 mg kg$^{-1}$) than in mudflat soil (5.74 ± 0. 30 mg kg$^{-1}$). The AK content in rhizosphere soils was 3.7%–18.7% lower than that in mudflat soil. The soils were ranked, from highest AS concentration to lowest, as follows: *K. obovata* rhizosphere soil (217.7 ± 5.5 mg kg$^{-1}$) > *S. alterniflora* rhizosphere soil (204.3 ± 5.1 mg kg$^{-1}$) > mudflat soil (184.3 ± 4.0 mg kg$^{-1}$) > *S. glauca* rhizosphere soil (166.7 ± 3.0 mg kg$^{-1}$) > *S. apetala* rhizosphere soil (156.0 ± 9.2 mg kg$^{-1}$) > *C. scabrifolia* rhizosphere soil (133.7 ± 6.1 mg kg$^{-1}$).

## Bacterial diversity and community composition

After filtering, 1,010,741 high-quality 16S rRNA gene sequences were obtained from 18 soil samples, with an average sequence length of 421 bp. All the rarefaction analysis curves were near saturation, indicating that the sequencing depth was sufficient to capture the biodiversity in the samples (Fig. 1A). Compared with mudflat soil, the *K. obovata* and *S. apetala* rhizosphere soils contained significantly more OTUs (higher Sobs index) and had higher species diversity (higher Shannon's index) ($P < 0.05$, Table 2). The other samples were ranked, from highest bacterial alpha diversity to lowest, as follows: *C. scabrifolia* rhizosphere soil > *S. alterniflora* rhizosphere soil > mudflat soil > *S. glauca* rhizosphere soil, but the differences were not significant ($P > 0.05$). The Ace species richness tended to be higher in rhizosphere soils than in mudflat soil ($P > 0.05$), with the highest values in the rhizosphere soils of *S. apetala*, *S. alterniflora*, and *K. obovata*. In general, mangrove restoration increased soil bacterial diversity to a greater degree than did *S. alterniflora* invasion. There were 2,049 OTUs common to all samples, 594 OTUs unique to mudflat soil, and 269 OTUs, 265 OTUs, 223 OTUs, 156 OTUs, and 123 OTUs unique to the rhizosphere soils of *K. obovata*, *S. alterniflora*, *S. apetala*, *C. scabrifolia*, and *S. glauca*, respectively (Fig. 1B).

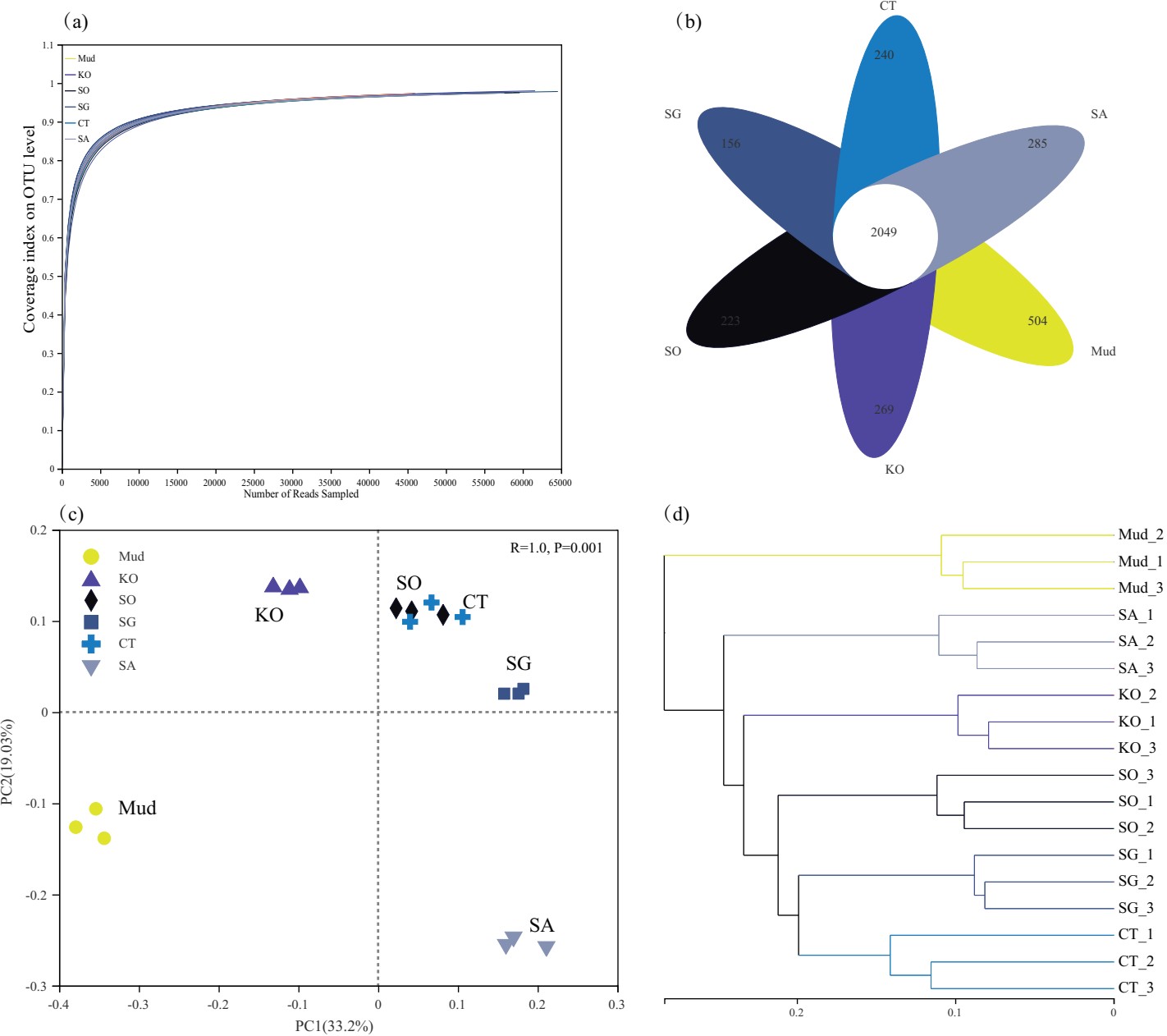

**Figure 1 Rarefaction curves of bacterial communities (A), Venn diagram showing unique and shared OTUs (B), principal component analysis (C) and unweighted pair-group method with arithmetic means (UPGMA) cluster analysis (D) at OUT levels in different soil samples.** Mud, bare mudflat; KO, *Kandelia obovata*; SO, *Sonneratia apetala*; SG, *Suaeda glauca*; CT, *Carex scabrifolia*; SA, *Spartina alterniflora*.

In terms of beta diversity, both the PCA and cluster analyses revealed a substantial effect of vegetation type on the rhizosphere bacterial community (Figs. 1C and 1D). In the PCA analysis, axes 1 and 2 accounted for 52.2% of the total variance in bacterial community composition (PERMANOVA, *P* < 0.01). The first PCA axis separated rhizosphere soil of *K. obovata* from mudflat soil and other rhizosphere soils, while the second axis separated rhizosphere soil of *S. alterniflora* from mudflat soil and the other rhizosphere soils

**Table 2 Alpha diversity indices of the bacterial community.**

| Site | Sobs | Shannon | Simpson | Ace |
|------|------|---------|---------|-----|
| Mud | 3,698 ± 26cd | 6.63 ± 0.04c | 0.0056 ± 0.0006a | 5,027 ± 15d |
| KO | 3,965 ± 122ab | 6.78 ± 0.09ab | 0.0042 ± 0.0005b | 5,458 ± 113ab |
| SO | 3,983 ± 100a | 6.80 ± 0.06a | 0.0043 ± 0.0006b | 5,688 ± 181a |
| SG | 3,587 ± 74d | 6.62 ± 0.05c | 0.0040 ± 0.0003b | 5,183 ± 115cd |
| CT | 3,823 ± 93bc | 6.80 ± 0.06a | 0.0030 ± 0.0003c | 5,275 ± 151bc |
| SA | 3,818 ± 40bc | 6.69 ± 0.03bc | 0.0039 ± 0.0003b | 5,526 ± 71a |

**Note:**
Mud, bare mudflat; KO, *Kandelia obovata*; SO, *Sonneratia apetala*; SG, *Suaeda glauca*; CT, *Carex scabrifolia*; SA, *Spartina alterniflora*; Different lowercase letters indicated the significant differences among different vegetation types (Tukey's test, *P* < 0.05).

(Fig. 1C). This result indicated that restoration with *K. obovata* and invasion of *S. alterniflora* significantly affected bacterial communities, but the effect of *K. obovata* was stronger (*P* < 0.05). The results of the PCA also revealed a high degree of similarity among the soil bacterial communities in rhizosphere soils of *C. scabrifolia*, *S. apetala*, and *S. glauca*. The cluster analysis clustered the soil samples into six groups, indicating that different types of vegetation resulted in the formation of distinct bacterial communities (Fig. 1D).

The OTUs obtained from all soil samples were classified into 58 phyla, 188 classes, 441 orders, 684 families, and 1,245 genera. The dominant bacterial phyla across all samples were Proteobacteria (26.85–36.98%), Actinobacteria (5.27–13.62%), Desulfobacterota (6.42–10.44%), and to a lesser extent Acidobacteria, Bacteroidetes, Chloroflexi, Myxococcota, and Firmicutes (Fig. 2A). As shown in Fig. 2B, the dominant bacterial classes across all samples were Gammaproteobacteria (15.77–27.38%), Alphaproteobacteria (6.31–14.17%), Bacteroidetes (4.07–13.31%), Acidimicrobiia (3.22–5.50%), Polyangia (2.34–6.54%), and Desulfuromonadia (2.06–5.39%). We performed a LEfSe analysis (LDA score > 4) to identify differentially abundant taxa among the rhizosphere soils and mudflat soil (Fig. 3). A total of 52 bacterial branches exhibited significant differences, with 9 and 43 differentially abundant taxa in the mudflat soil and rhizosphere soils, respectively. More precisely, the LEfSe analysis identified 7, 6, 11, 6, and 14 differentially abundant clades or taxa in the rhizosphere soils of *K. obovata*, *S. apetala*, *S. glauca*, *C. scabrifolia*, and *S. alterniflora*, respectively. The phylum Proteobacteria and class Alphaproteobacteria were enriched in the rhizosphere soil of *S. alterniflora*, while the phylum Desulfobacterota, class Bacilli, and order Exiguobacterales were enriched in the rhizosphere soil of *K. obovata*. The rhizosphere soil of *C. scabrifolia* was enriched with the phyla Firmicutes and Myxococcota and the class Polyangia.

## Relationships between soil properties and bacterial communities

First, soil properties, such as pH, EC, OM, AN, *etc.*, were retained *via* backward selection based on the VIF (VIF < 10). Then, a distance-based redundancy analysis (Bray–Curtis metrics) was used to explore the relationships between bacterial community structure and soil characteristics. As shown in Fig. 4, all the soil variables accounted for 50.85% of the

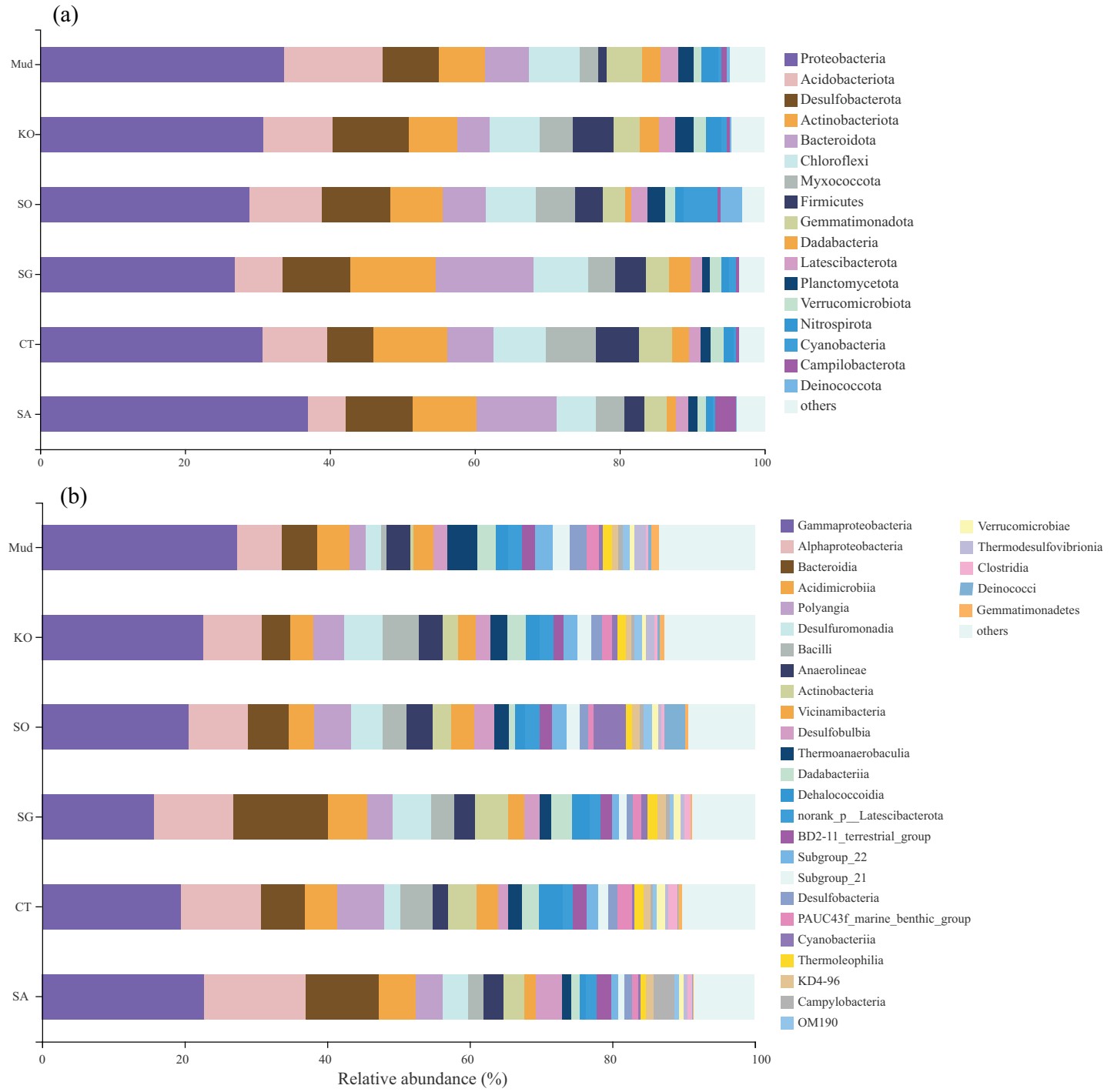

**Figure 2 The proportion of the bacterial phyla (A) and class (B) of six soil samples.** Mud, bare mudflat; KO, *Kandelia obovata*; SO, *Sonneratia apetala*; SG, *Suaeda glauca*; CT, *Carex tristachya*; SA, *Spartina alterniflora*.

variance, with axis 1 explaining 34.15% of the variance and axis 2 explaining another 16.70%. The major soil properties driving soil bacterial community composition were EC ($R^2 = 0.57$, $P < 0.01$), AK ($R^2 = 0.52$, $P < 0.01$), AP ($R^2 = 0.49$, $P < 0.05$), and OM ($R^2 = 0.36$, $P < 0.05$). Among them, soil EC and AK were the most significant factors determining

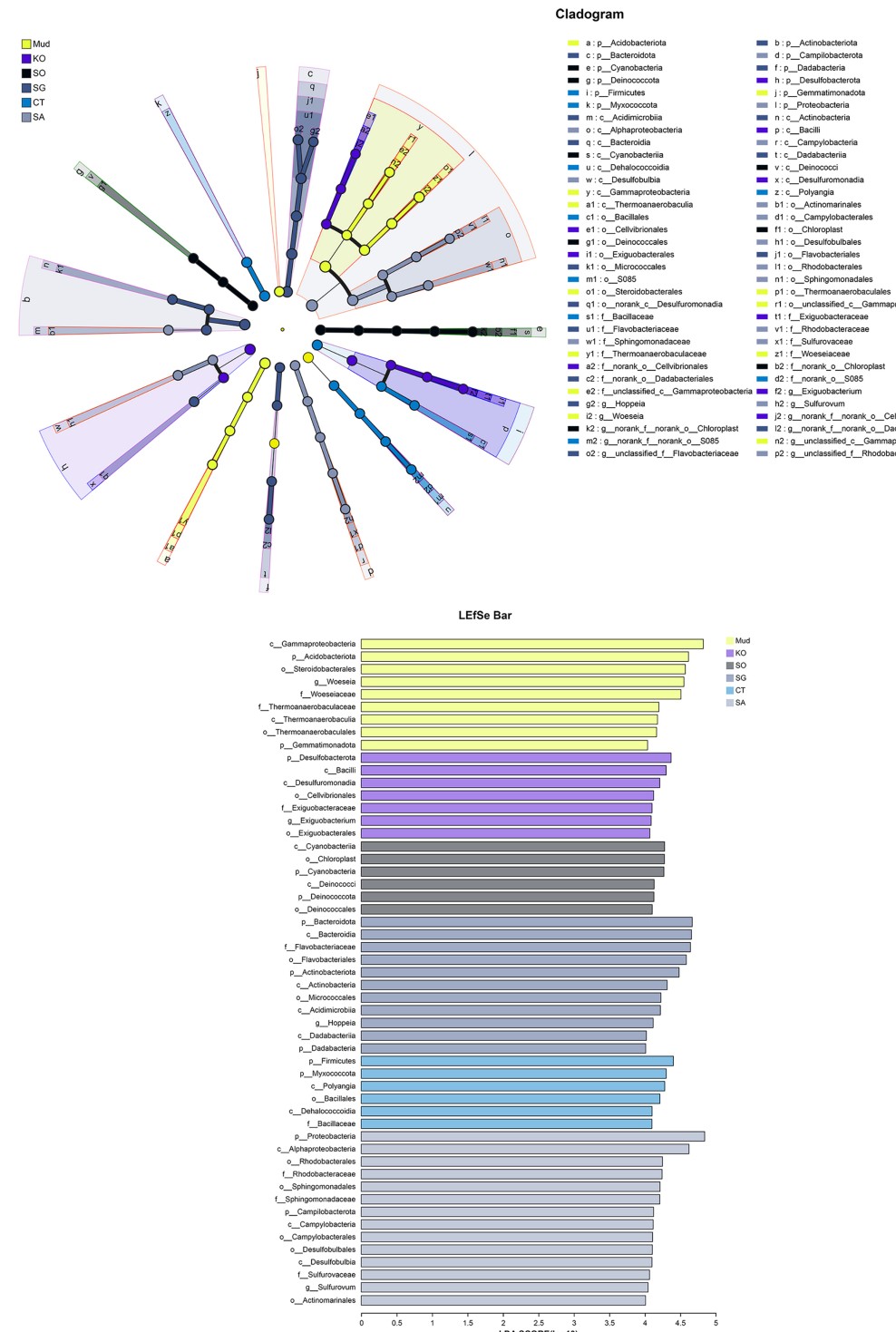

**Figure 3 Different taxon (from phylum to genus) response to the treatments based on Lefse analysis.** Mud, bare mudflat; KO, *Kandelia obovata*; SO, *Sonneratia apetala*; SG, *Suaeda glauca*; CT, *Carex tristachya*; SA, *Spartina alterniflora*.               

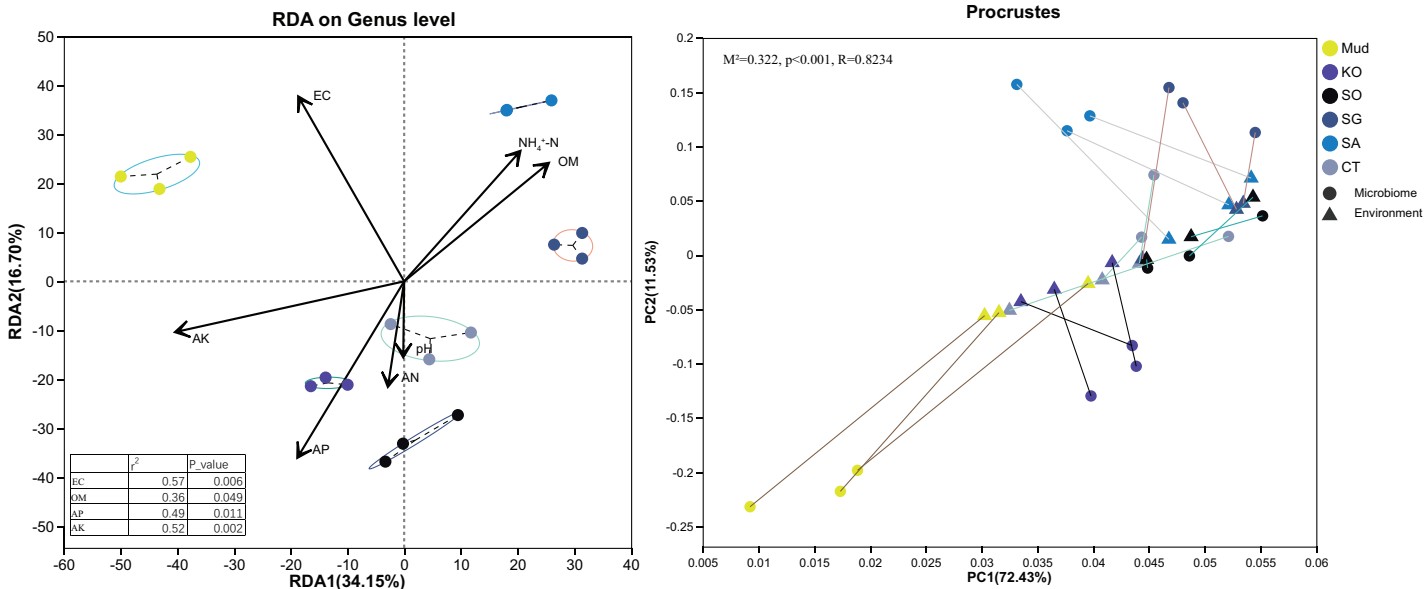

**Figure 4 Redundancy analysis and Procrustes analysis of environmental factors and soil bacterial community structure.** Physicochemical vector correlation plots showing strengths and directions of the relationships between the physicochemical variables. Axis legends representing the percentages of variation explained by the axis. Mud, bare mudflat; KO, *Kandelia obovata*; SO, *Sonneratia apetala*; SG, *Suaeda glauca*; CT, *Carex tristachya*; SA, *Spartina alterniflora*.

bacterial community composition in the rhizosphere soils of native *C. scabrifolia* and *S. glauca*, respectively. Soil AP and OM significantly affected the bacterial communities in the rhizosphere soils of *K. obovata* and *S. alterniflora*, respectively. The results of the Procrustes analysis confirmed the very significant consistency between the spatial ranking of bacterial community structure and soil physicochemical properties ($P < 0.01$).

## Potential functional roles of bacterial communities

According to the PCA results (Fig. 1C), the KEGG functional profiles of bacterial communities from mudflat soil and rhizosphere soils of *K. obovata*, *S. alterniflora*, and *C. scabrifolia* were predicted using PICRUSt2. As shown in Fig. 5, the LEfSe analysis (LA > 2.5) showed that soil bacteria metabolic functions related to anabolism, such as biosynthesis of secondary metabolites (ko01110), biosynthesis of amino acids (ko01230), aminoacyl-tRNA biosynthesis (ko00970), lipopolysaccharide biosynthesis (ko00540), folate biosynthesis (ko00790), and protothenate and CoA biosynthesis (ko00770), were enriched in mudflat soil. Some pathways related to environmental adaption, including oxidative phosphorylation (ko00190), bacterial secretion system (ko03070), carbon (C) fixation pathways in prokarytes (ko00720), ribosome (ko03010), homologous recombination (ko03440), and protein export (ko03060), were also enriched in mudflat soil. In the rhizosphere soils, bacterial catabolic pathways related to energy metabolism were significantly enriched and varied among the different vegetation types. Specifically, glycine, serine and threonine metabolism (ko00260), valine, leucine and isoleucine degradation (ko00280), benzoate degradation (ko00362), and butanoate metabolism (ko00650) were enriched in the rhizosphere soil of native *C. scabrifolia*. Starch and sucrose

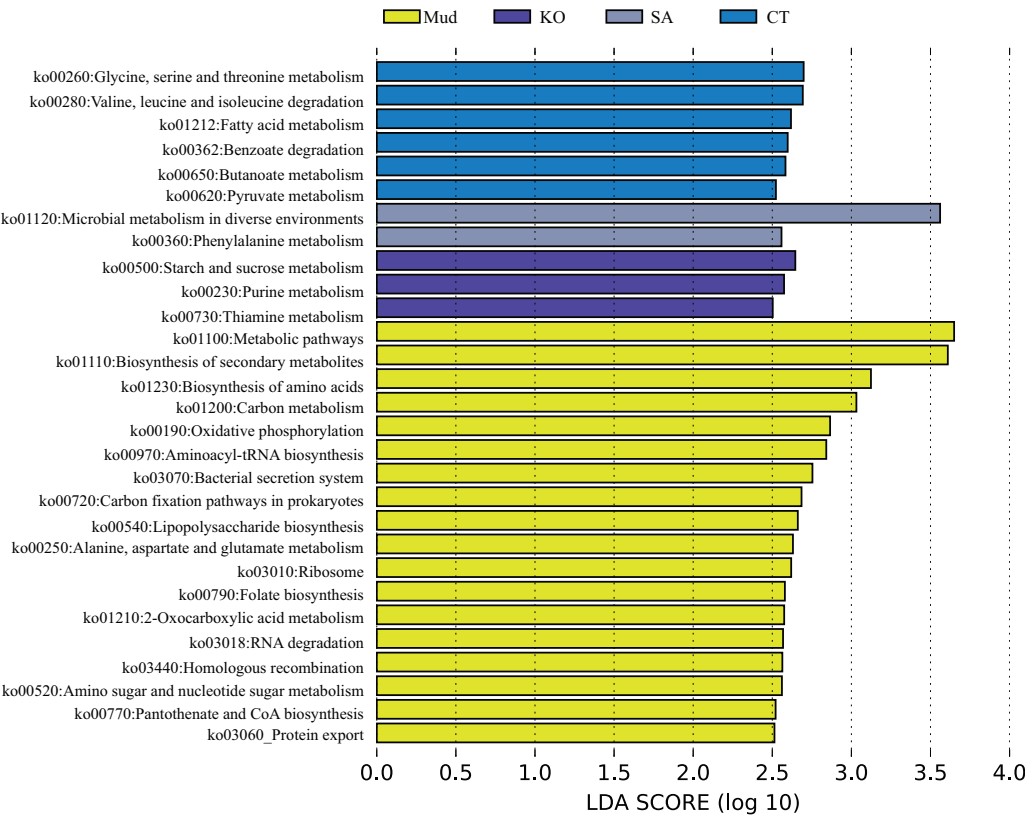

**Figure 5 The metagenomic function was predicted by PICRUSt2.** Identification of crucial rhizosphere soil bacterial KEGG pathway associated with different vegetation types used LEfSe analysis. Mud, bare mudflat; KO, *Kandelia obovata*; CT, *Carex tristachya*; SA, *Spartina alterniflora*.

metabolism (ko00500), purine metabolism (ko00230), and thiamine metabolism (ko00730) were enriched in the rhizosphere soil of the mangrove *K. obovata*, while microbial metabolism in diverse environments (ko01120) and phenylalanine metabolism (ko00360) were enriched in the rhizosphere soil of invasive *S. alterniflora*. In summary, the predicted functions of rhizosphere soil bacteria included active catabolism related to nutrient cycling and metabolic pathways that varied among vegetation types, whereas the predicted functions of mudflat soil bacteria were related to synthesis and resistance to the external environment.

## DISCUSSION

Plant growth can effectively improve coastal saline soil. The rhizosphere is particularly important in this process because it facilitates salt adsorption, it contains root exudates that fuel microorganisms and organic acids that neutralize soil pH, and it harbors fine roots that eventually decompose to produce humic, fulvic, and carbonic acids (*Centenaro et al., 2018*; *Liu et al., 2022b*). In our study, the EC was generally lower in rhizosphere soils than in mudflat soil, but was significantly higher in *S. alterniflora* rhizosphere soil than in all other soil samples (Table 1). This finding was consistent with those of *Yang et al. (2013, 2020)*, who documented higher soil salinity and higher soil bacterial abundance and

diversity in *S. alterniflora* rhizosphere soil than in soil samples from a bare flat and native salt marsh communities (*e.g.*, *S. salsa*, *S. mariqueter*, and *P. australis*). *Song et al. (2023)* found that soil salinity was relatively higher in *S. alterniflora* rhizosphere soil than in mudflat soil and rhizosphere soil of the mangrove *K. obovata*. Thus, we speculated that the ability to increase soil salinity may be an efficient strategy by which *S. alterniflora* successfully invades coastal areas.

Generally, high salinity increases the osmotic potential of soil, restricts the growth of heterotrophic bacteria, and ultimately leads to decreased microbial biomass. However, in this study, we detected higher bacterial diversity in the *S. alterniflora* rhizosphere soil (with higher EC) than in mudflat soil (with lower EC) (Table 2). A possible reason for this phenomenon might be that the promoting effects of the increased OM content offset the inhibitory effects of high salinity on bacterial growth in the *S. alterniflora* rhizosphere soil. In contrast to most native herbs, *S. alterniflora* has a consistently high net photosynthetic rate, and its invasion can markedly increase the nutrient levels in local ecosystems in coastal areas by generating plant residues (*i.e.*, litter and roots) (*Yang et al., 2020*). It has been estimated that 10–50% of C assimilated by higher plants *via* photosynthesis is released by roots as exudates, which provide nutrients or function as signaling molecules to boost microbial growth and development (*Hinsinger, Plassard & Jaillard, 2006*). In support of this, we found that more OM accumulated in the *S. alterniflora* rhizosphere soil than in the other soils ($P < 0.05$, Table 1). The high OM content regulated bacterial community succession (Fig. 4), and resulted in increased bacterial species richness (Table 2).

Bacteroidetes are known to degrade complex organic compounds and hydrolyze polysaccharides such as cellulose and starch (*Fernández-Gómez et al., 2013*). Previous studies have shown that *S. alterniflora* residues contain large amounts of recalcitrant substances with a high C/N ratio (*Zhang et al., 2020*; *Yang et al., 2020*), which partly explains the dynamics of Bacteroidetes in coastal wetlands after *S. alterniflora* invasion. *Yang et al. (2020)* also detected increased abundance of Bacteroidetes following *S. alterniflora* invasion in a coastal area, accompanied by a reduction in the accumulation of organic C and N in soils. In addition, *S. alterniflora* was shown to recruit some specific microbial taxa associated with dissimilatory nitrate reduction (*Gao et al., 2019*). This may explain the higher levels of AN and $NO_3^-$-N in *S. alterniflora* rhizosphere soil than in the other soils in our study ($P < 0.05$, Table 1). We found that Desulfobulbales was enriched in the soil bacterial community after *S. alterniflora* invasion (Fig. 3), and this may have promoted dissimilatory S reduction (*Song et al., 2022*). *Wang et al. (2019)* also found that S storage in soil was higher in a coastal area invaded by *S. alterniflora* than in areas dominated by native plant communities, consistent with the higher AS content in the *S. alterniflora* rhizosphere soil than in mudflat soil ($204.3 \pm 5.1$ mg kg$^{-1}$ *vs*. $184.3 \pm 4.0$ mg kg$^{-1}$, Table 1) in our study. The high S content may decrease soil pH through the oxidation of sulphides and the production of sulfuric acid. Consistent with this, the *S. alterniflora* rhizosphere soil had the lowest pH ($8.39 \pm 0.09$) among all the soil samples in this study (Table 1). The Sphingomonadales includes iron (III) -reducing bacteria that are involved in C and N cycling (*Peng et al., 2016*), and this taxon was enriched in the soil bacterial community after *S. alterniflora* invasion. Further research is needed to explore
biogeochemical processes involving iron, and the relationships with C, N, and S cycling, to comprehensively assess the invasion mechanisms of *S. alterniflora*.

Mangrove afforestation is widely used to replace invasive *S. alterniflora* for ecological restoration of coastal wetlands (*Zhao et al., 2016*). Because mangroves are woody halophytes, their exudates and litter residues release more available nutrients for use by microbes than do *S. alterniflora* and other native saltmarsh plants. It has been reported that the amount of litter produced by mangroves equates to approximately 33% of their net primary production, with mean values ranging from 114 to 1,700 g m$^{-2}$ yr$^{-1}$ (*Shi et al., 2024*). As shown in Table 1, soil AP, OM, AN, and NO$_3^-$-N contents were typically higher in the mangrove rhizosphere soils than in other soils, suggesting that restoration with mangroves increased the soil nutrient pool, as well as contributing to habitat reconstruction. Plants with a high biomass are often associated with greater soil microbial abundance and diversity (*Steinauer, Chatzinotas & Eisenhauer, 2016*). Thus, the highest abundance, species richness, and diversity of soil bacteria were in the *K. obovata* rhizosphere soil (Table 2). Consistent with this result, previous studies have also reported that restoration with mangroves can increase the abundance and diversity of microbial communities, and promote nutrient cycling (*Lu et al., 2022*; *Gomes et al., 2010*; *Hu et al., 2024*). It has been reported that P plays an important role in the growth and productivity of mangrove ecosystems (*Krauss et al., 2008*). Most of the inorganic P present in coastal soil is bound to calcium, iron, and aluminum ions as insoluble P. *Reddy et al. (2021)* detected bioavailable P limitation in mangrove ecosystems in five different areas along the east and west coasts of India. It has been reported that Bacilli (*Thatoi et al., 2013*) and Exiguobacteraceae (*Bharti et al., 2013*) in the mangrove rhizosphere exhibit high P activity, and release soluble P into the soil solution. In anaerobic soil in the mangrove ecosystem, Desulfobacterota usually function as the main decomposers of OM and play a role in the mineralization of organic S and in the production of soluble iron and P (*Holguin, Vazquez & Bashan, 2001*). Consistently, we found that Bacilli, Exiguobacteraceae, and Desulfobacterota were enriched in the *K. obovata* rhizosphere soil compared with other soils (Fig. 3). Moreover, the soil AP content in *K. obovata* rhizosphere soil (11.77 ± 1.48 mg kg$^{-1}$) was higher than that in other soils (range, 4.73 ± 0.11 to 6.41 ± 0.32 mg kg$^{-1}$). The results of the RDA further indicated that AP was the most important environmental factor driving bacterial community succession in the *K. obovata* rhizosphere (Fig. 4). Another study found that, as the AP concentration in soil increased over the chronosequence of mangrove restoration, there were increases in the abundance of P-related genes, including those involved in organic P mineralization (*phnX/W*, *phoA/D/G*, *phnJ/N/P*), inorganic P solubilization (*gcd*, *ppx-gppA*), and P transport (*phnC/D/E*, *pstA/B/C/S*) (*Hu et al., 2024*). *Yan et al. (2021)* reported that high contents of nucleic acid P and metabolite P might increase the cold tolerance of *K. obovata*. Similarly, *Shi et al. (2024)* also found that a high leaf P concentration in winter may be part of the adaptive response of *K. obovata* to low-temperature stress, which is more severe in high-latitude regions. Correspondingly, the *K. obovata* individuals in the artificial mangrove plantations in the northernmost part of the study area may need to accumulate P to adapt to seasonal freezing and occasional cold-temperature events. Additionally, it should be noted that the lowest soil AP content

was in the *S. alterniflora* rhizosphere. These findings imply that certain management practices, such as the addition of P fertilizer, could improve the growth and survival of mangroves at higher latitudes, thereby enhancing their effectiveness in controlling invasive *S. alterniflora*. Generally, planting *K. obovata* is an effective method to control invasive *S. alterniflora* in coastal areas, because it not only increases soil microbial abundance, species richness, and diversity, but also attracts P-solubilizing bacteria, such as Bacilli, Exiguobacteraceae, and Desulfobacterota, to promote P cycling.

Different plant species release diverse sets of small organic compounds that change rhizosphere conditions and support high levels of microbial activity (*Hinsinger, Plassard & Jaillard, 2006*). Thus, nutrient substrates are considered to be strong drivers of rhizosphere soil bacterial community structure, diversity, and function. However, we found that EC was the most important factor regulating bacterial community succession in the *C. scabrifolia* rhizosphere soil (Fig. 4), which had the lowest EC value of $3.74 \pm 0.23$ ds m$^{-1}$. Similarly, *Morrissey et al. (2014)* found that soil salinity was closely associated with bacterial community structure in tidal wetlands. *Jiang et al. (2023)* reported that salinity decreased the abundance of N- and P-related microorganisms in an intertidal wetland through osmotic stress and toxic effects. In mangrove soil, a strong negative correlation was detected between salinity and the abundance of denitrifying genes, such as *nirK* and *nosZ* (*Wang et al., 2018*). In our study, a LEfSe analysis revealed the remarkable enrichment of Firmicutes and Bacilli in rhizosphere soil of *C. scabrifolia*, relative to other soils (Fig. 3). *Nicholson & Fathepure (2005)* and *Cheng, Chen & Zhang (2018)* noted that Firmicutes have bioremediation functions in hypersaline conditions. Thus, the increased abundance of Firmicutes in the *C. scabrifolia* rhizosphere could be conducive to the improvement of salinized soil. *Sui et al. (2023)* studied bacterial communities in a tropical mangrove nature reserve, and reported that Firmicutes could withstand long periods of immersion in seawater in the intertidal zone by forming spores. Because of this survival advantage, it was the second most abundant phylum. The KEGG functional profiling analysis indicated that the *C. scabrifolia* rhizosphere soil bacteria had mainly active catabolic functions, while the mudflat bacterial community, which was enriched with some oligotrophic bacteria, including Acidobacteriota and Gemmatimonadota, had functions related to synthesis and resistance to environmental factors (Fig. 3). In summary, compared with *S. alterniflora* and *K. obovata*, native *C. scabrifolia* improved coastal saline soil primarily through decreasing EC rather than by increasing nutrient levels, and thus its effect on promoting bacterial richness and diversity was weaker than that of mangroves.

## CONCLUSIONS

This study is the first attempt to reveal the effects of *S. alterniflora* invasion and consequent mangrove afforestation on the structures of bacterial communities in soil. Our results show that soil EC, AK, AP, and OM are the main environmental factors driving bacterial community composition in the studied wetland. Briefly, *S. alterniflora* accumulated OM and increased the EC, and recruited copiotrophic Bacteroidetes to its rhizosphere. All of these changes contributed to its invasiveness. Mangroves, especially *K. obovata*, were found to be suitable pioneer trees for coastal restoration, and recruited beneficial

Desulfobacterota and Bacilli in their rhizospheres. This ultimately increased the contents of AP, AS, and AN in soil. Among these soil factors, AP was not only the most important factor driving bacterial community succession, but also might increase the cold tolerance of *K. obovata* at high latitudes. These results emphasize the strong effect of mangrove afforestation to increase soil nutrient contents and subsequently shape the bacterial communities and their associated functions. Our results provide a reference for designing effective ecological restoration strategies, and for the sustainable development of coastal biodiversity and functionality. We note that our results are based only on 16S rRNA gene sequences, so further in-depth investigations including metagenomic sequencing, identification of specific functional genes, and metabolomic analyses should be conducted to explore the complexity of soil–microbiome–plant interactions and to comprehensively assess the effects of introducing mangroves into intertidal ecosystems.

### Funding
This study was supported by the Zhejiang Provincial Natural Science Foundation of China under Grant (No. LGF22C030003), the People's Government of Zhejiang Province and Chinese Academic of forestry (No. 2022SY08), and the "Pioneer" and "Leading Goose" R&D Program of Zhejiang (No. 2024C02002). The funders had no role in study design, data collection and analysis, decision to publish, or preparation of the manuscript.

### Grant Disclosures
The following grant information was disclosed by the authors:
Zhejiang Provincial Natural Science Foundation of China: LGF22C030003.
People's Government of Zhejiang Province and Chinese Academic of Forestry: 2022SY08.
"Pioneer" and "Leading Goose" R&D Program of Zhejiang: 2024C02002.

### Competing Interests
The authors declare that they have no competing interests.

### Author Contributions
- Jinwang Wang conceived and designed the experiments, prepared figures and/or tables, authored or reviewed drafts of the article, and approved the final draft.
- Xi Lin conceived and designed the experiments, performed the experiments, prepared figures and/or tables, authored or reviewed drafts of the article, and approved the final draft.
- Xia An conceived and designed the experiments, analyzed the data, prepared figures and/or tables, authored or reviewed drafts of the article, and approved the final draft.
- Shuangshuang Liu analyzed the data, prepared figures and/or tables, authored or reviewed drafts of the article, and approved the final draft.
- Xin Wei analyzed the data, prepared figures and/or tables, authored or reviewed drafts of the article, and approved the final draft.
- Tianpei Zhou conceived and designed the experiments, prepared figures and/or tables, authored or reviewed drafts of the article, and approved the final draft.
- Qianchen Li performed the experiments, prepared figures and/or tables, authored or reviewed drafts of the article, and approved the final draft.
- Qiuxia Chen performed the experiments, prepared figures and/or tables, authored or reviewed drafts of the article, and approved the final draft.
- Xing Liu conceived and designed the experiments, performed the experiments, analyzed the data, prepared figures and/or tables, authored or reviewed drafts of the article, and approved the final draft.

## Data Availability

All raw sequencing data are available at NCBI Sequence Read Archive (SRA) database: SRP500226.

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
