# Peer review of "Mangrove afforestation as an ecological control of invasive Spartina alterniflora affects rhizosphere soil physicochemical properties and bacterial community in a subtropical tidal estuarine wetland"

_PeerJ, doi:10.7717/peerj.18291_

## Round 0.1 · original submission · Major Revisions

Dear authors, As you can see from the reviews, the topic of your manuscript has been considered interesting. However, the reviewers raised several important issues. Please try to address them all. All the issues raised by reviewer 1 should be addressed and the suggestions taken into account.

·

Basic reporting

No comment

Experimental design

Experimental design was appropriate to answer the research question. However, more replications would be needed to make statistical models powerful to observe treatment effects.

Validity of the findings

No comment

Additional comments

I enjoyed reading this manuscript. The authors have interesting data showing how S. alterniflora invasion and planting of mangroves as ecological control affect soil chemical properties and soil microbial communities.
Here are the major issues to improve the manuscript:
1. Line 183: I don't think three replicates are powerful enough statistically to capture variation. Did you test for homogeneity of variance and normality?
2. Line 230: Use broken line to show the sequencing depth on the rarefaction curve. Why did you choose 9604 as sequencing depth? From your curve, it seemed you could have gone deeper.
3. The discussion can be strengthened and easier to flow if the authors can focus on discussing their results using their data and making it flow logical. For example, the first two paragraphs can focus on your most important results. Talk about how the different plant communities affect soil chemical properties, focusing on how S alterniflora affects EC and OM and how this is related to soil microbial communities observed in your study. Use your data to support your discussion. Then go ahead to discuss how planting mangroves of ecological control modify the effects of the invasive S. alterniflora. Also remember, that correlation does not always imply causation.
Other minor issues include:
1. Title: I suggest that you rewrite this title to reflect how your introduction is structured and the actual work. It could be written as "Mangrove afforestation as an ecological control of invasive spartina alterniflora affects rhizopheric ..... "
2. Line 25: Please include another sentence to connect the previous sentence on mangroves and Spartina alterniflora, with root-associated microbes in the next sentence.
3. Line 67: Remove the word “the” from the beginning of the sentence.
4. Line 95: From your method description, the S alterniflora invasion occurred first and mangroves were planted as ecological control. Can you please reword this sentence accordingly? Except you are talking able the S. alterniflora invasion of the planted mangroves themselves.
5. Line 97: change “drive” to “driving”
6. Line 103: change “does” to “do”
7. Line 112: remove the word “typical” from the sentence.
8. Line 116: I suggest you complete the classification. Is the soil Histosol?
9. Line 134: This sentence sounds incomplete. Please correct it.
10. Line 228: include space between “samples” and “with”
11. Line 240: remove the word “to”
12. Line 261: Change to “brief” to “briefly”.
13. Line 273: What does significantly positive mean? Please describe.

Reviewer 2 ·

Basic reporting

The paper explored the soil physicochemical properties and bacterial community structure in rhizosphere of Spartina alterniflora, mangroves, and native plants in the a estuarine wetland. A lot of previous studies had quantified the effects of Spartina alterniflora inversion on soil bacterial and fungal community. Therefore, the novelty of current study is enough.
If the author focused on rhizosphere soil, which is decoupled with the topic of plant inversion. On one hand, the author could answer more novel questions that how Spartina alterniflora inversion affect on the rhizosphere bacterial community of native plants. However, I did not see this information.
On the other hand, the author could focus on the influence of species on the bacterial community in rhizosphere soil. From this perspective, several plant traits should be provided, such as root and litter traits.

Experimental design

The current experimental design is unrelated the background of Spartina alterniflora inversion. Actually, authors studied the influence of plant species. Therefore, more novel information associated with how plant species regulate bacterial community may provide novelty.

Validity of the findings

Different plant species holding different bacterial community is expected, the author did not provide mechanisms for such difference. Some direct evidences, such as root and litter traits, is needed.

---

## Round 0.2 · Minor Revisions

Dear authors, Your manuscript has significantly improved by the review process. But, as you will see in the attached review, there are still some minor issues that have to be addressed.

Kind regards
Elisabeth Grohmann

·

Basic reporting

I appreciate the review of the discussion section by the authors based on my comments. It flows better and it is easier to understand.

Many grammatical errors were introduced due to the new review. Please reread and ensure that every sentence is grammatically correct. The authors can employ the help of a professional editing service to improve the readability of the manuscript.

Experimental design

After changing the sequencing depth for rarefaction from 9604 to 41216, the authors did not mention if they reanalyzed the alpha and beta diversity. I believe these parameters will be affected since they are mostly influenced by rarefaction sequencing depth. If the authors did these and no changes (which I doubt) were observed, they should mention it.

Please include your approach to testing for homogeneity of variance and normality in the section, data analysis.

Please include the reference for soil classification. Instead of saying “the soil is belonging to Inceptisol”, why not say “the soil is classified as an Inceptisol”

Validity of the findings

No comment

---

## Round 0.3 · accepted · Accept

The authors carefully revised the manuscript. It can now be accepted for publication in the journal. The manuscript now meets the criteria for acceptance by the journal.